# Investigation of Bacterial Isolations and Antimicrobial Susceptibility of Chronic Rhinitis in Cats

**DOI:** 10.3390/ani12121572

**Published:** 2022-06-17

**Authors:** Wannisa Meepoo, Tassanee Jaroensong, Chantima Pruksakorn, Jatuporn Rattanasrisomporn

**Affiliations:** 1Kasetsart University Veterinary Teaching Hospital, Faculty of Veterinary Medicine, Kamphaeng Saen Campus, Kasetsart University, Nakhon Pathom 73140, Thailand; kungowon@gmail.com; 2Department of Companion Animal Clinical Sciences, Faculty of Veterinary Medicine, Kasetsart University, 50 Ngamwongwan Rd., Bangkok 10900, Thailand; fvettsj@ku.ac.th; 3Department of Veterinary Microbiology and Immunology, Faculty of Veterinary Medicine, Kasetsart University, 50 Ngamwongwan Rd., Bangkok 10900, Thailand; 4Faculty of Veterinary Medicine, Kasetsart University Veterinary Teaching Hospital, 50 Ngamwongwan Rd., Bangkok 10900, Thailand; 5Center for Advanced Studies for Agriculture and Food, Kasetsart University Institute for Advanced Studies, Kasetsart University, Bangkok 10900, Thailand

**Keywords:** rhinitis, cats, bacteria, antimicrobial susceptibility

## Abstract

**Simple Summary:**

Although rhinitis is a quite common disease in cats, the underlying cause is not easily explored. In addition to responding to the emerging concern regarding antimicrobial resistance and close contact between a pet and its owner, this study investigated bacterial identification and antimicrobial susceptibility in chronic rhinitis in cats. The current study found that rhinitis was more likely in younger and young adult cats, with *Pseudomonas* spp. being the main bacterial species, as well as multidrug-resistant, followed by *Pasteurella* spp., *Staphylococcus* spp., and *E. coli*. Overall, amikacin, imipenem, and meropenem, which are intravenous antimicrobial agents, exhibited high activity against most bacterial species. However, the use of antimicrobials in categories A and B (EMA) in companion animals are not recommended if other antimicrobial choices are available. Notably, this report reflects on the current antimicrobial resistance situation. Therefore, the appropriate antimicrobial usage and selected drugs should be based on antimicrobial use guidelines, the result of culture, and antimicrobial susceptibility.

**Abstract:**

Chronic rhinitis is a quite common upper respiratory tract (URT) disease in cats. As a result of unclear etiology, frequently, multidrug-resistant bacteria are identified. This study investigated bacterial isolations and an antimicrobial susceptibility test (AST) in chronic rhinitis in cats. The medical records of 395 cats with chronic URT signs were reviewed at the Kasetsart University Veterinary Teaching Hospital (KUVTH) between 2016 and 2021 to survey the underlying causes of URT. Then, apart from rhinitis, other causes were excluded to identify the bacterial species and antimicrobial susceptibility. The results indicated that the most frequent finding was neoplasia, followed by rhinitis and anatomical defects. Furthermore, the only significant association was between the age range and disease group, with gender, FIV, or FeLV infection not being significant. Rhinitis was 4.7 times more likely to occur than neoplasia in younger and young adult cats in the age range < 1–3 years compared to the group > 10 years. The main bacterial species was the *Pseudomonas* species. Antimicrobials with a susceptibility rate of more than 90% were amikacin, gentamicin, ciprofloxacin, norfloxacin, marbofloxacin, imipenem, and meropenem. In conclusion, rhinitis was the second most common chronic URT disease in cats and was more common in younger and young adult cats. The predominant bacteria with AST in this study reflect the antimicrobial resistance situation. Thus, antimicrobial usage should follow antimicrobial use guidelines first.

## 1. Introduction

Nowadays, chronic rhinitis is a quite common cause of upper respiratory tract (URT) diseases in cats [1,2]. Approximately 50% of rhinitis cases fail to identify the underlying causes [3]. For that reason, rhinitis is difficult to treat and is usually associated with multidrug-resistant bacteria. Studies have reported *Pasteurella multocida* [4], *Pseudomonas* spp. [3], *Escherichia coli*, *Staphylococcus* spp. [4], *Streptococcus* spp. [4], *Bordetella bronchiseptica* [5], and *Mycoplasma* spp. [6] as being common in feline chronic rhinitis. A recent study revealed that the nasal cavity in cats consists of a diverse bacterial microbiome [7] where commensal, pathogenic bacteria were inhabitants. The correlations between nasal microbiome and diseases in cats were not determined. However, they were reviewed in human studies which mentioned that dysbiosis of the microbiome can lead to the overgrowth of opportunistic bacteria and cause pathogenic conditions [8]. Moreover, the bacterial composition can be affected by multiple factors such as aging, diseases, antibiotic treatment, feeding type, etc. [8,9,10,11]. The evidence of bacteria being a primary cause of chronic rhinitis in cats was rare [2]. One study presumed primary bacterial rhinosinusitis-associated optic neuritis in a cat [12] that involved *E. coli* and *Actinomyces* spp. Although bacterial identification in the nasal cavity was frequently found in mixed cultures of commensal germs, an organism found with heavy pure growth would be considered to be pathogenic [2].

Since antimicrobial resistance (AMR) can occur via horizontal gene transfer and gene mutation, inappropriate antimicrobials usage in cats with chronic rhinitis can exacerbate it by resistant selection in successive bacterial generations which have acquired AMR ability to other antimicrobials [13]. Furthermore, the emergence of AMR transmission between pets and humans has increased over the last decade, with examples being methicillin-resistant *S. pseudintermedius* (MRSP) and extended-spectrum beta-lactamase (ESBL)-production *E. coli* [14]. As a consequence of the current close relationships between companion animals and their owners, the transmission of resistant bacteria to humans could increase either by direct contact (licking or physical injury) or environmental contamination [10]. For example, domestic cats having close contact with their owners are at a significantly higher risk of *Staphylococcus aureus* colonization and acquired antimicrobial resistant determinants [15]. Antimicrobial resistance is one of the top issues of concern worldwide not only because morbidity and mortality rates are rising, but also because therapeutic costs and the length of hospital stays are increasing, particularly in children, the elderly, and immunocompromised patients. In humans, HIV (human immunodeficiency virus) infection has been reported to cause CD4+ destruction, leading to immune dysfunction that can result in developing an opportunistic infection, increasing the risk of cancer, and enhancing the prevalence of allergic rhinitis [16]. Similarly, FIV (feline immunodeficiency virus)/FeLV (feline leukemia virus) infection in cats can cause immune dysfunction similar to HIV in humans and has been reported in oncogenesis involvement. However, there is still no evidence of a relationship to rhinitis.

Antimicrobial usage and identifying resistant bacteria are the keys to possibly slowing antimicrobial resistance and preventing the spread of infection. Thus, the current study aimed to investigate bacterial identification and antimicrobial susceptibility in feline chronic rhinitis and find the association between retroviral infection affecting cats’ immunity and URT diseases. Eventually, other etiology of upper respiratory tract disease will be identified, as well.

## 2. Materials and Methods

### 2.1. Animal Experiments and Clinical Evaluation

This research was a retrospective study that reviewed the medical records of cats that had been examined at the Rhinoscopic Clinic of the Kasetsart University Veterinary Teaching Hospital (KUVTH) between 2016 and 2021. Medical records were inspected to obtain information on signalment, clinical signs, FIV and FeLV testing (Witness^®^ FeLV/FIV, Zoetis, USA), the results of cytology, and the histopathology of nasal biopsy specimens, bacterial cultures, and antimicrobial susceptibility testing. In total, 395 cats presenting with upper respiratory signs for more than 3 weeks were included and surveyed for the etiology of URT diseases.

Then, cats with specific causes except for rhinitis, such as anatomical defects (nasopharyngeal stenosis), polyps, neoplasia, and fungal infection were excluded to determine bacterial isolation and drug sensitivity in chronic rhinitis cases. In addition, cats with rhinitis that had not had bacterial identification and an antimicrobial susceptibility test were excluded. The results of the bacterial culture and antimicrobial susceptibility tests were collected from the rhinoscopic specimens that were sent to the Microbiological Unit of the KUVTH. Nasal deep tissues were collected using rhinoscopy and sent to the KUVTH laboratory for cytology or histopathology. In addition, nasal flush samples having significantly higher bacterial cultures than nasal tissue were examined for aerobic bacterial isolation and antimicrobial susceptibility [17]. The specimens were cultured on blood agar and MacConkey agar under aerobic conditions at 37 °C for 18–24 h. Culture plate were examined for the types of colonies and classified as pure or mixed culture. Bacteria were primary identified based on colony morphology, Gram’s staining, catalase and oxidase tests. Further bacterial identification was performed by standard biochemical methods. Antimicrobial susceptibility testing were performed using the conventional method (agar disc diffusion) according to the Clinical and Laboratory Standards Institute from animals (M31-A3, CLSI, 2008). The tested antimicrobials were amikacin (30 µg), amoxicillin (10 µg), amoxicillin-clavulanic acid (30 µg), azithromycin (15 µg), ceftriaxone (30 µg), cephalexin (30 µg), ciprofloxacin (5 µg), clindamycin (2 µg), doxycycline (30 µg), enrofloxacin (5 µg), gentamicin (10 µg), metronidazole (5 µg), imipenem (10 µg), marbofloxacin (5 µg), norfloxacin (10 µg), sulfa-trimethoprim (25 µg), and meropenem (10 µg). The quality control strains, *Escherichia coli* ATCC 25922, *S. aureus* ATCC 25923, and *P. aeruginosa* ATCC 27853, were included. The zone of inhibition from an antimicrobial agent of interest was classified into one of three categories: susceptible, intermediate, or resistant.

### 2.2. Statistical Analysis

Descriptive analyses of the etiology of URT diseases and bacterial identifications with antimicrobial susceptibility were summarized as percentages. A chi-square test was used to identify differences in age, sex, and retroviral testing for each etiology of URT diseases. If relevance was found, multinomial logistic regression was applied to assess the magnitude of association. Statistical analysis was performed using the SPSS statistical software and a *p*-value of < 0.05 was considered to be significant.

## 3. Results

### 3.1. Overall Chronic Upper Respiratory Tract Disease in Cats

In total, 395 cats had upper respiratory signs consisting of neoplasia (48.1%), rhinitis (36.7%), anatomical defects (8.4%), fungal infection (4.8%), and other causes, such as polyps and foreign bodies (2%). Domestic shorthair (DSH) was the most common breed in this study, with 333 (84.3%) out of the 395 cats, followed by Persians (6.8%) and Scottish folds (2.8%), respectively.

The clinical signs of each chronic URT disease in cats, such as lymphadenopathy, epistaxis, dyspnea, facial deformity, sneezing, breathing noise, chronic nasal discharge, and nasal obstruction are presented in Figure 1. Epistaxis and facial deformity were notable in the neoplasia group, whereas breathing noise was most common in the miscellaneous group. Chronic nasal discharge was higher in all disease groups, especially in the rhinitis group, where purulent nasal discharge accounted for 62.0%.

As presented in the data above, apart from rhinitis and neoplasia, the other causes of URT, such as anatomical defects, polyps, and foreign bodies, were included in the miscellaneous group because of their small proportions. The ages of the cats were categorized in age ranges consisting of less than 1–3 years (31.9%), 4–6 years (21.3%), 7–10 years (29.9%), and more than 10 years (16.9%). The number of female cats was 49.9% and males represented 50.1%. In total, 180 cats (16.1%) subjected to the FIV test had a positive result, whereas 200 cats (27.5%) subjected to the FeLV test had a positive result (Table 1). A chi-square test was performed to find the relevance of URT disease with factors such as age range, sex, and retroviral infection. The only significant association was between age range and disease group, with gender, FIV, or FeLV infection not being significant (Table 2). After multinomial regression was run to evaluate the magnitude of association, rhinitis was 4.78 times more likely than neoplasia to occur in younger cats in the age range < 1–3 years compared to the > 10 years group. In addition, the miscellaneous causes (such as anatomical defects, polyps, and foreign bodies) were 1.53 times more common than neoplasia in younger and young adult cats. Furthermore, the miscellaneous group was 3.25 times more likely to have than rhinitis in the youngest age range compared to the > 10 years age group.

### 3.2. Bacterial Isolations from Cats with Rhinitis

After excluding the other causes, 109 cats met the criteria (rhinitis result and bacterial isolation, as well as antimicrobial susceptibility) which found 74.3% pure culture, whereas 25.7% was mixed culture. In addition, eight cats were found with no bacterial growth. *Pseudomonas* spp. was the main bacterial species (32.1%), followed by *Pasteurella* spp. (24.4%), *Staphylococcus* spp. (17.6%), *E. coli* (8.4%), *Klebsiella* spp. (5.3%), *Streptococcus* spp. (4.5%), *Enterobacter* spp. (2.4%), *Bacillus cereus* (2.3%), *Acinetobacter* spp. (1.4%), *Enterococcus* spp. (0.8%), and *Proteus mirabilis* (0.8%), respectively.

### 3.3. Antimicrobial Susceptibility

*Pseudomonas* spp. was found in about one-third (32.1%) of chronic rhinitis cases in cats. The antimicrobial agents showing a susceptibility rate of more than 90% were amikacin (95.8%), ciprofloxacin (95.1%), norfloxacin (95.1%), gentamicin (95.1%), imipenem (94.6%), marbofloxacin (94.3%), and meropenem (91.7%), as shown in Figure 2. In contrast, clindamycin and metronidazole were 100% resistant, followed by cephalexin (97.6%), amoxicillin or amoxicillin-clavulanic acid (95.1%), and sulfa-trimethoprim (90.2%), respectively. The second most common bacterial species was *Pasteurella* spp. (24.4%), and almost all antimicrobial agents were more than 90% susceptible, except for clindamycin, which was highly resistant (84.4%), as shown in Figure 3. While *Staphylococcus* spp. was found in 17.6% of cats with rhinitis, additionally, only two antimicrobial agents were more than 90% susceptible, namely, amikacin (100%) and imipenem (95.2%), as shown in Figure 4. Finally, *E. coli* was found to be the fourth most common in cats with rhinitis at 8.4%. Intravenous antimicrobial agents (amikacin, imipenem, and meropenem) were 100% susceptible, whereas the other agents had a susceptibility of less than 85% (Figure 5). Clindamycin and amoxicillin had 100% resistance.

## 4. Discussion

There has been increasing concern regarding antimicrobial resistance and the close relationship between companion animals and owners. Furthermore, rhinitis in cats is challenging to diagnose and treat, and as a result, developing multidrug-resistant organisms is easy. Therefore, the current study surveyed the underlying causes of rhinitis cats with chronic URT signs, and it also investigated bacterial identifications and antimicrobial susceptibility testing in cat with rhinitis. The results indicated that rhinitis was the second most common cause of chronic upper respiratory tract disease in cats, which has also been reported in other studies [18,19]. However, some evidence in other studies has indicated that rhinitis was the most common cause [1]. This might depend on the diversity of the diagnostic modality of the sample. Domestic shorthair (DSH) cats appeared to be the most common breed in this study.

Chronic nasal discharge was presented at higher levels in all chronic URT diseases, particularly purulent nasal discharge, which was found in approximately 62% of cats with rhinitis. This might indicate the presence of a bacterial infection, although viral or fungal agents could be involved as well [20]. Evidence of bacterial infection in cats with rhinitis has not yet been verified; however, several human studies have indicated that the nasal microbiome could play a crucial role as a modulator of the localized immune response, with dysbiosis of nasal microbiota being associated with a pathologic condition in the respiratory tract [9]. Furthermore, there are various factors that could affect nasal microbial composition, such as antibiotic treatment, birth mode, feeding method, smoking exposure, and location [8,11,21].

In the current study, rhinitis was more common in younger and young adult cats, as is the case for humans, where allergic rhinitis is common in childhood and adolescence [22]. In addition, allergic rhinitis was still decreasing with age for unclear reasons, and also had a high correlation with atopy, allergy, and asthma [23]. However, there is less animal-based evidence [1] to support these links and the etiology is still doubtful. Nevertheless, rhinitis in older humans has been reported in conjunction with physiological changes when aging, such as an increase in cholinergic activity, leading to mucosal gland and collagen fiber atrophy and impaired mucociliary function as the consequences [24].

Although some reports mentioned a retrovirus as having either a direct or indirect effect on oncogenesis [25,26], the current study revealed no significant association between retroviral infection in all disease groups. However, the current study used the Zoetis witness^®^ test kit to detect the antigens of FeLV and antibodies of FIV in cats. Even though their sensitivity and specificity are more than 90%, the results could be non-detectable, particularly in FeLV cats with regressive infection. The current study also found that gender was unlikely to relate to disease groups, similar to other studies [1,18].

Several studies mentioned that FHV-1 might be the primary cause of chronic rhinitis as a result of mucosal epithelium and turbinate destruction that could lead to secondary bacterial infection. However, there was no difference reported in FHV-1 isolation between cats with rhinitis and healthy cats [6]. In addition, FHV infection has a latent period that is not easy to diagnose [27]. *Bordetella bronchiseptica*, *Chlamydophila felis*, *Mycoplasma* spp., and the anaerobic bacteria that have been mentioned as respiratory pathogens in other research were not evident in the current study. *Chlamydophila felis* could not survive for prolonged periods of time outside the host, and diagnosing feline chlamydiosis requires the use of specialized culture techniques [28]. Furthermore, *Mycoplasma* spp. is difficult to culture compared to other bacteria and requires special media, with several weeks required for some colonization results to be evident [29]. Additionally, to detect *Bordetella bronchiseptica* using bacterial culture, it is necessary to use a selective medium to reduce other bacteria overgrowth [5].

The current study revealed that the main bacterial species in chronic rhinitis in cats was *Pseudomonas* spp., which corresponded with another report [3]. However, this result was inconsistent with other recent studies which found that *Pasteurella* spp. was the most common [4,7]. Since *Pseudomonas* spp. is commonly found in samples from the deeper nasal tissue of cats [2,6], the difference might have been due to sample collection procedures. Some evidence showed that *Pseudomonas* spp. was more abundant in the nasal communities of people as well, especially in hospitalized groups [11]. Additionally, the proportion of microbial communities was also influenced by the ventilation in the environment. Interestingly, the current study revealed that only amikacin, imipenem, and meropenem (which are parenteral antimicrobial agents) and fluoroquinolones (except enrofloxacin) exhibited high activity for *Pseudomonas* spp. Notably, most cats with chronic upper respiratory tract diseases had received several antimicrobials before they were referred to KUVTH. These might encourage antimicrobial resistance as a consequence. However, the data on ciprofloxacin and norfloxacin usage in cats are limited in the veterinary literature. Since *Pseudomonas* spp. is a multidrug-resistant organism and antimicrobials seem to be ineffective, performing nasal flushing under anesthesia to remove the loculated *Pseudomonas* spp. has been reported to have some advantages [20]. The next most common bacterial species was *Pasteurella* spp., which is part of the commensal of URT in cats and also causes lower respiratory tract infection [30]. Furthermore, commensals can change into pathogens if they are under continuous pressure from macrophages [31] or triggered by localization, such as the anatomical site and the microenvironment [32]. *Pasteurella* species in the current study showed high susceptibility to nearly all antimicrobials, apart from clindamycin. This result of high susceptibility in *Pasteurella* spp. is similar to the latest report [33]. Nevertheless, this result contrasts with a recent study [4] that indicated *Pasteurella multocida* was the most common infectious agent in cats and only pradofloxacin was effective against it. The reason for the geographic variation is unclear, but it is often related to patterns of antimicrobial usage.

*Staphylococcus* spp. and *E. coli* in the current study were found in small proportions. Interestingly, the current report showed that these bacterial species were more than 90% susceptible to parenteral antimicrobials only, unlike the results have been reported in other studies, which showed low levels of AMR in both bacterial species [33].

Because it is a retrospective study, the current study has limitations. Data, including the minimal inhibitory concentration and anaerobic bacteria, are not available because the aerobic bacterial culture and disc diffusion methods are routinely investigated in the KUVTH laboratory, and some bacteria which require selective media are not detected. Further, there was no repeat testing of retroviruses to clarify infection status. Finally, bacterial species in healthy cats are not investigated, and thus it is difficult to tell significantly of bacterial findings, as shown in another report [7].

This report reflects the current antimicrobial resistance situation. Importantly, other underlying causes of chronic URT should first be excluded before treating bacterial infection. However, this report does not recommended the use of antimicrobials in categories A and B such as carbapenems in companion animals if other antimicrobial choices are available, in accordance with the European guidelines [34], which indicate the use of antibiotics of categories D and C in veterinary medicine and preserve the use of categories A and B for human medicine. Accordingly, the selection of drugs should follow antimicrobial use guidelines [20] and cultures with an antimicrobial susceptibility test.

## 5. Conclusions

Rhinitis is the second most common chronic URT disease in cats and is more common in younger and young adult cats. *Pseudomonas* spp. was the main bacterial species, followed by *Pasteurella* spp., *Staphylococcus* spp., and *E. coli*. Overall, only amikacin, imipenem, and meropenem had high susceptibility, while clindamycin was less susceptible to all the investigated bacterial species. Importantly, these results reflect the current antimicrobials resistance situation. Thus, antimicrobial usage should follow antimicrobial use guidelines and the results of cultures and an antimicrobial susceptibility test.

## Figures and Tables

**Figure 1 animals-12-01572-f001:**
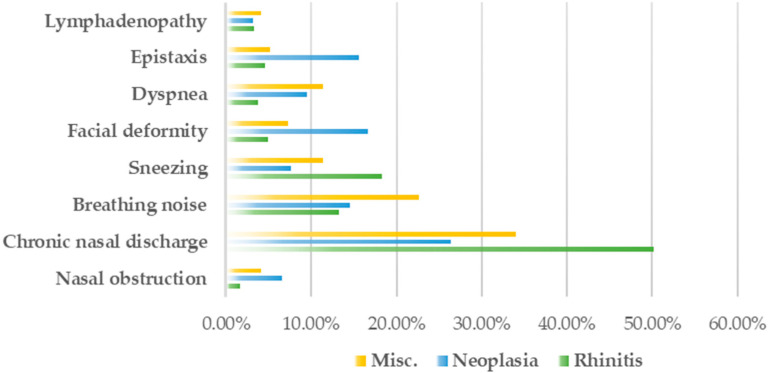
Clinical signs of each chronic URT disease in cats (Misc. includes anatomical defects, polyps, and foreign bodies).

**Figure 2 animals-12-01572-f002:**
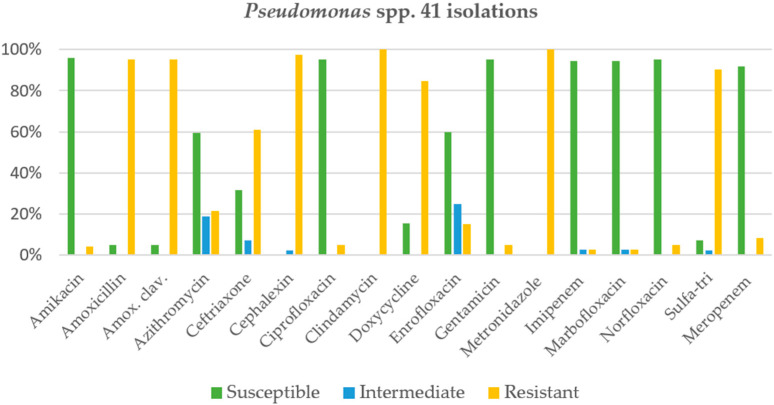
Activity of antimicrobials against *Pseudomonas* spp. in 41 isolations.

**Figure 3 animals-12-01572-f003:**
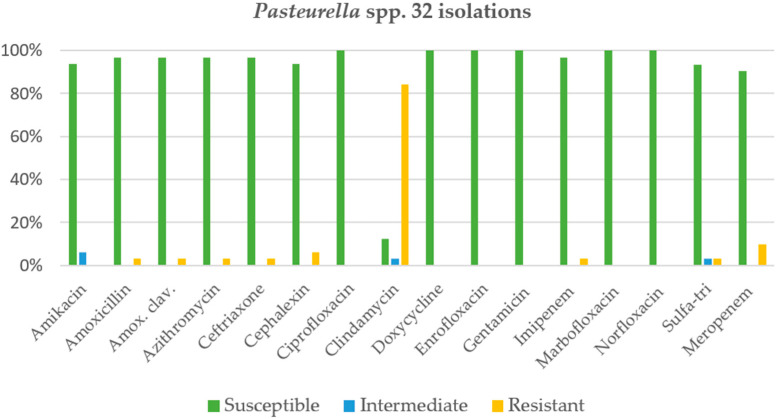
Activity of antimicrobials against *Pasteurella* spp. in 32 isolations.

**Figure 4 animals-12-01572-f004:**
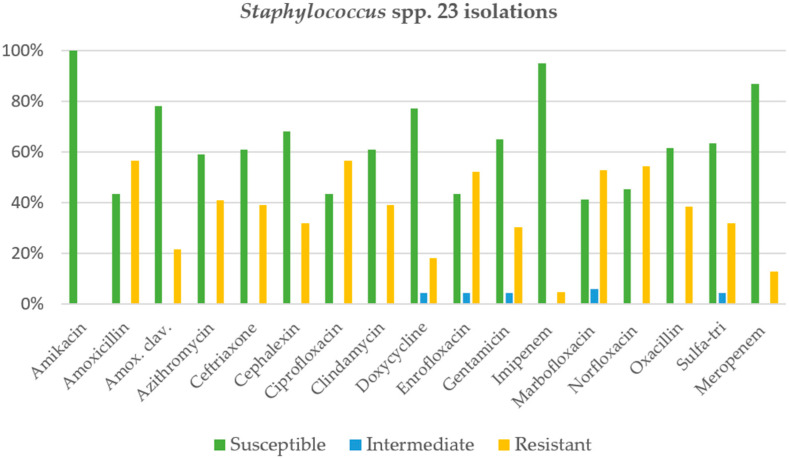
Activity of antimicrobials against *Staphylococcus* spp. in 23 isolations.

**Figure 5 animals-12-01572-f005:**
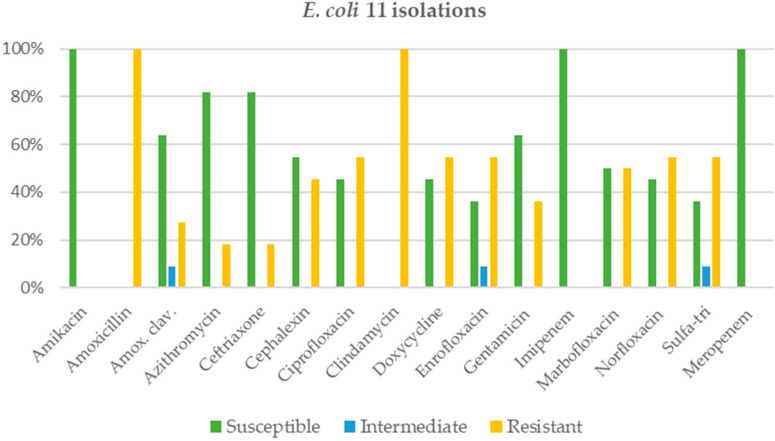
Activity of antimicrobials against *E. coli* in 11 isolations.

**Table 1 animals-12-01572-t001:** Frequency of each predisposing factor across disease groups.

Factor	Disease Group			Total	*p*-Value
	Rhinitis	Neoplasia	Misc.		
Age range					<0.001
<1–3 years	60 (47.6%)	27 (21.4%)	39 (31.0%)	126 (31.9%)	
4–6 years	34 (40.5%)	42 (50.0%)	8 (9.5%)	84 (21.3%)	
7–10 years	31 (26.3%)	78 (66.1%)	9 (7.6%)	118 (29.9%)	
>10 years	20 (29.9%)	43 (64.2%)	4 (5.9%)	67 (16.9%)	
Gender					0.077
Female	64 (32.5%)	106 (53.8%)	27 (13.7%)	197 (49.9%)	
Male	81 (40.9%)	84 (42.4%)	33 (16.7%)	198 (50.1%)	
FIV					0.108
Positive	7 (24.1%)	18 (62.1%)	4 (13.8%)	29 (16.1%)	
Negative	65 (43.0%)	63 (41.7%)	23 (15.3%)	151 (83.9%)	
FeLV					0.062
Positive	14 (25.5%)	32 (58.2%)	9 (16.3%)	55 (27.5%)	
Negative	63 (43.4%)	62 (42.8%)	20 (13.8%)	145 (72.5%)	

Misc. includes fungal infection, anatomical defects, and others.

**Table 2 animals-12-01572-t002:** Coefficients and odds of chronic URT causes.

Variable	Rhinitis vs. Neoplasia		Misc. vs. Rhinitis		Misc. vs. Neoplasia	
	B	Exp (B)	B	Exp (B)	B	Exp (B)
Age groups (ref > 10 years)						
<1–3 years	1.564	4.778 **	1.179	3.250 **	2.743	15.528 **
4–6 years	0.554	1.740	0.163	1.176	0.717	2.048
7–10 years	−0.157	0.854	0.373	1.452	0.215	1.240

Chi-square = 70.101; D.f. = 6; Sig. (**) = 0.000; Cox and Snell R^2^; Misc. includes fungal infection, anatomical defects, and others, e.g., polyps and foreign bodies.

## Data Availability

The data presented in this study are available on request from the corresponding author.

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
