# Peer review of "Investigation of Bacterial Isolations and Antimicrobial Susceptibility of Chronic Rhinitis in Cats"

_animals, 2022, doi:10.3390/ani12121572_

Round 1
Reviewer 1 Report
I suggest to avoid the use of personal form (i.e. our, we…) throughout the text.
Throughout the text several sentences are redundant. Please check and delete the repetition.
The title well reflects the main aim and findings of the work.
The abstract adequately summarize results and significance of the study. However, Authors should rewrote the sentence describing the study’s goals “This study investigated bacterial pathogen and antimicrobial susceptibility in chronic rhinitis in cats, as well as identifying the underlying cause of chronic URT disease and determining the association between retroviral infection and URT disease.” as it is too long and results unclear.
The introduction section is well written and it falls within the topic of the study, and Authors cited appropriately bibliographic information.
The section of Materials and Methods is clear for the reader and it meticulously describes the methods applied in the study. However some information should be added.
What about the inclusion/exclusion criteria for the enrolment of animals?
.
Regarding statistical analysis, Did Authors perform a normality test in order to test the normal distribution of data?
Authors wrote “Bacterial identification and antimicrobial susceptibility testing were performed using the conventional method (agar disc diffusion)…” more detailed information regarding this aspect of methodology is needed.
Results section as well as Discussion section is clear and well written, the findings obtained in the study were well discussed and justified with appropriate references.
However, considering that there are 7 tables and 2 figures, I suggest to simplify the results section.
Tables are generally good and well represents the findings of the study.
The conclusion section should be rewritten more clearly. Authors should better summarize the results and the significance of the study.
Authors should check and standardize the references in the list according to journal guidelines.
Author Response
Response to reviewer 1 comments
Point 1: I suggest to avoid the use of personal form (i.e. our, we…) throughout the text.
Throughout the text several sentences are redundant. Please check and delete the repetition.
Response 1: Thank you for your suggestion. I have corrected redundant sentences.
Point 2: The abstract adequately summarize results and significance of the study. However, Authors should rewrote the sentence describing the study’s goals “This study investigated bacterial pathogen and antimicrobial susceptibility in chronic rhinitis in cats, as well as identifying the underlying cause of chronic URT disease and determining the association between retroviral infection and URT disease.” as it is too long and results unclear.
Response 2: I have rewritten as your suggestion.
Point 3: The section of Materials and Methods is clear for the reader and it meticulously describes the methods applied in the study. However some information should be added.
What about the inclusion/exclusion criteria for the enrolment of animals?
Response 3: The author had added inclusion/exclusion criteria in Materials and Methods already in line 102, 106-107.
Inclusion criteria is cats with presenting upper respiratory signs more than 3 weeks at Rhinoscopy clinic.
Exclusion criteria is cats with results of rhinitis from rhinoscopic cytology or histopathology are not perform bacterial identification and antimicrobial susceptibility test.
Point 4: Regarding statistical analysis, Did Authors perform a normality test in order to test the normal distribution of data?
Response 4: The author did not perform normality test since the data in our study are categorical scales such as Age range, sex, breed.
Point 5: Authors wrote “Bacterial identification and antimicrobial susceptibility testing were performed using the conventional method (agar disc diffusion)…” more detailed information regarding this aspect of methodology is needed
Response 5: Thank you for your suggestion. I have added the detail in line 116-122.
Point 6: considering that there are 7 tables and 2 figures, I suggest to simplify the results section.
Response 6: Some tables and figures were cut off already.
Point 7: The conclusion section should be rewritten more clearly. Authors should better summarize the results and the significance of the study.
Response 7: I have rewritten already.
Point 8: Authors should check and standardize the references in the list according to journal guidelines.
Response 8: The reference style was downloaded from the journal instruction. However, I have corrected as your suggestion.

Reviewer 2 Report
Line 25- 28, 41- 44: authors seems to imply that antimicrobials of high importance (amikacin, imipenem, meropenem) should be used in treating rhinitis in cats. This is a dangerous idea because most bacterial rhinitis is secondary to an underlying disease.
Line 53- 62: author talked about bacteria found in cat rhinitis and then started talking about antimicrobial resistance in human which does not seem to flow and relate to each other. Author then jumped back to talk about Pasteurella spp. in cat upper respiratory tract infection in line 63. Can author elaborate on the role of bacteria in chronic rhinitis in cats and discuss the background and evidence of bacteria being a primary cause of chronic rhinitis and how antimicrobial resistance might be develop in cats?
Line 74- 82: Yes, antimicrobial resistance is a concerning issue but how does this relate to bacteria in cats' nose? How does bacteria from cats' nose cause antimicrobial resistance in people? There is evidence of Staphylococcus aureus colonization in cats' owners and other bacteria but this does not seem to relate to the bacteria in cats' nose?
Line 105: Can authors please describe how neoplasia, polyps, fungal infection are ruled out? Has CT been performed in these animals? Has nasal foreign body been ruled out? Can authors please describe the workup process in details?
Figure 2: This figure does not highlight much major finding in this paper. Would authors consider not including this figure but to describe this finding?
Table 3: Can authors please put the exact p value for age range? If not possible, please put "p<0.001"?
Line 87 and discussion part (line 287): The word "etiology" of upper respiratory disease used in line 87 is confusing and misunderstanding to readers. While it is fine to discuss the association between chronic rhinitis and bacteria identified in the study, it is dangerous to put a causative link between the two and make readers think that chronic rhinitis is directly linked/ primarily caused by bacterial infection. In most cases of chronic rhinitis, bacterial infection is secondary. You have indicated on line 295 that the evidence of bacterial infection in rhinitis cats has not yet been verified, however this seems to be sentences throughout the whole paper implying that bacterial infection is a primary cause of chronic rhinitis using wordings like "etiology" or "causes".
The nasal cavity is also not a sterile region and finding bacteria in the nasal cavity is not unexpected (ie. there is bacterial microbiome in nose of healthy cats). Since there is no control group (healthy cats) in this study, it is hard to know whether the differences in bacterial species found in this study is related to chronic rhinitis or regional differences in Thailand. Therefore, the conclusion of Pseudomonas being the main pathogen in chronic rhinitis in cats cannot be made (line 331) And the use of the word "pathogen" also means that authors think this organism is the cause of disease to the host (ie. this is the definition of pathogen). This is again, a dangerous concept as bacteria is often not the primary cause of chronic rhinitis in cats. Authors need to change the word to "bacterial species" rather than using the word "pathogen".
Line 281- 283: Can author clarify why development of resistant organisms is easy? How is this related to rhinitis being hard to treat and diagnose?
Line 313- 315: I agree that these are limitations of the study. Can author put all the limitations in one paragraph at the last part of the discussion please?
Line 325 - 330: these are also limitations. Can author put these limitations all in one paragraph?
Line 334- 338: as discussed above, because there is no control group of this study, it is difficult to know if the differences in bacterial species found in this study (compared to other studies such as Dorn et al 2017) is from regional differences or chronic rhinitis. You have also explained a few other reasons in line 334- 338 which are valid. But a big limitation here (which authors need to address) is lack of control groups.
Line 338- 341: Initially, authors discuss about the importance and concerns to antimicrobial resistance in human medicine. However, the discussion in line 338- 341 almost imply that readers should use these antimicrobials (of high importance (amikacin, imipenem and meropenem) to treat bacterial rhinitis in cats. This is an extremely dangerous idea and will further contribute to antimicrobial resistance in both human and veterinary medicine. It is not unsurprising that authors found Pseudomonas spp. being resistant to most antimicrobials apart from these high importance ones, however authors should be cautious when discussing these findings that underlying nasal disease should exclude first by extensive workup before treating bacterial infection and these antimicrobials are antimicrobials of high importance and should not be used easily. Authors should also refer readers to the ISCAID guidelines (Lappin 2017) with use of antimicrobials.
Reference 33: please use another more appropriate referencing, this is for urinary tract infections.
Author Response
Response to reviewer 2 comments
Point 1: Line 25- 28, 41- 44: authors seems to imply that antimicrobials of high importance (amikacin, imipenem, meropenem) should be used in treating rhinitis in cats. This is a dangerous idea because most bacterial rhinitis is secondary to an underlying disease.
Response 1: Actually, this study would like to make people realize the AMR situation in animal health care these days. Antimicrobial usage should follow antimicrobial use guidelines (Lappin et al., 2017) and European medicines agency guidelines. However, I have rewritten as it makes misunderstanding.
Point 2: Line 53- 62: author talked about bacteria found in cat rhinitis and then started talking about antimicrobial resistance in human which does not seem to flow and relate to each other. Author then jumped back to talk about Pasteurella spp. in cat upper respiratory tract infection in line 63. Can author elaborate on the role of bacteria in chronic rhinitis in cats and discuss the background and evidence of bacteria being a primary cause of chronic rhinitis and how antimicrobial resistance might be develop in cats?
Response 2:
-Since, the nasal cavity of cats consists of diverse bacterial microbiome (Dorn, 2017) where commensal and potential pathogenic bacteria were inhabiting. Although, it is still unclear about the association between nasal microbiome and diseases in cats. The effect of nasal microbiome changing link to immune system and infectious disease was reviewed in human study “The microbiome of the nose -friend or foe?” (Dimitri-Pinheiro et al., 2020), ”The microbiome of the upper respiratory tract in health and disease” (Kumpitsch et al., 2019). Which involves local immune modulatory. Dysbiosis of nasal microbiome can lead to pathogenic conditions (Rawls, 2019).
-Bacteria being a primary cause of chronic rhinitis were rare (Ferguson et al., 2019) (Reed, 2020). One study (Moghaddam et al., 2020) presumed primary bacterial rhinosinusitis-associated optic neuritis in a cat that involves E. coli and Actinomyces spp. Reed, 2020 mentioned frequency of bacterial identification was mixed growth. However, one organism which found heavy pure growth would consider to be pathogenic.
-AMR can occur via horizontal gene transfer and gene mutation. Thus, inappropriate antimicrobials usage in cats with chronic rhinitis can exacerbate it by resistant selection in successive bacterial generations to increase their ability to acquire other antimicrobials (Reyaert, 2018).
Point 3: Line 74- 82: Yes, antimicrobial resistance is a concerning issue but how does this relate to bacteria in cats' nose? How does bacteria from cats' nose cause antimicrobial resistance in people? There is evidence of Staphylococcus aureus colonization in cats' owners and other bacteria but this does not seem to relate to the bacteria in cats' nose?
Response 3: In my view, antimicrobials usage in cats with chronic rhinitis is not affected only the nose but the entire body which is covered with a diverse microbiome. Inappropriate antimicrobials usage in companion animals encourages selection pressure which increases resistant bacteria in the animal microbiome and possibility spread to humans (European medicine agency; EMA). Thus, veterinarians can slow antibiotic resistance by selecting appropriate drugs following antimicrobial use guidelines and EMA that preserve antibiotics for animals and people. Transmission of AMR bacteria between animals and people could occur in several ways such as direct contact by licking, biting, and scratching or indirectly by contaminating food and the environment (Damborg et al., 2016) (Bhat,2021).
Point 4: Line 105: Can authors please describe how neoplasia, polyps, fungal infection are ruled out? Has CT been performed in these animals? Has nasal foreign body been ruled out? Can authors please describe the workup process in details?
Response 4: - Neoplasia, polyps, foreign bodies, and fungal infection were role out by rhinoscopy and histopathologic results. Cats with suspicious neoplasia were performed CT scan. This retrospective study was reviewed on medical records and found that generally work up process begins with history taking, physical examination, laboratory analysis (hematology, biochemistry, urinalysis, FIV & FeLV infection), diagnostic testing which starts at the least to the most invasive in chronic diseases such as cytology, bacterial culture & antimicrobial susceptibility test, radiographs, CT, Rhinoscopy with histopathology.
Point 5: Figure 2: This figure does not highlight much major finding in this paper. Would authors consider not including this figure but to describe this finding?
Response 5: Thank you for your suggestion.
Point 6: Table 3: Can authors please put the exact p value for age range? If not possible, please put "p<0.001"?
Response 6: I could not find the exact p value for age range. Thus, I put p<0.001 instead as your suggestion.
Point 7: Line 87 and discussion part (line 287): The word "etiology" of upper respiratory disease used in line 87 is confusing and misunderstanding to readers. While it is fine to discuss the association between chronic rhinitis and bacteria identified in the study, it is dangerous to put a causative link between the two and make readers think that chronic rhinitis is directly linked/ primarily caused by bacterial infection. In most cases of chronic rhinitis, bacterial infection is secondary. You have indicated on line 295 that the evidence of bacterial infection in rhinitis cats has not yet been verified, however this seems to be sentences throughout the whole paper implying that bacterial infection is a primary cause of chronic rhinitis using wordings like "etiology" or "causes".
Response 7: I have rewritten the wording.
Point 8: The nasal cavity is also not a sterile region and finding bacteria in the nasal cavity is not unexpected (ie. there is bacterial microbiome in nose of healthy cats). Since there is no control group (healthy cats) in this study, it is hard to know whether the differences in bacterial species found in this study is related to chronic rhinitis or regional differences in Thailand. Therefore, the conclusion of Pseudomonas being the main pathogen in chronic rhinitis in cats cannot be made (line 331) And the use of the word "pathogen" also means that authors think this organism is the cause of disease to the host (ie. this is the definition of pathogen). This is again, a dangerous concept as bacteria is often not the primary cause of chronic rhinitis in cats. Authors need to change the word to "bacterial species" rather than using the word "pathogen".
Response 8: That is right. I have corrected that as your suggestion.
Point 9: Line 281- 283: Can author clarify why development of resistant organisms is easy? How is this related to rhinitis being hard to treat and diagnose?
Response 9: Although chronic rhinitis in cats was done in all diagnostic processes. Some cats with chronic rhinitis are still doubted the underlying cause. These would affect the remedy’s efficacy. As a result, inappropriate antimicrobials usage will encourage resistant germs.
Point 10: Line 313- 315: I agree that these are limitations of the study. Can author put all the limitations in one paragraph at the last part of the discussion please?
Line 325 - 330: these are also limitations. Can author put these limitations all in one paragraph?
Response 10: Thank you for your suggestion. I have added limitations as your advice.
Point 11: Line 334- 338: as discussed above, because there is no control group of this study, it is difficult to know if the differences in bacterial species found in this study (compared to other studies such as Dorn et al 2017) is from regional differences or chronic rhinitis. You have also explained a few other reasons in line 334- 338 which are valid. But a big limitation here (which authors need to address) is lack of control groups.
Response 11: Thank you for your suggestion. I have added it.
Point 12: Line 338- 341: Initially, authors discuss about the importance and concerns to antimicrobial resistance in human medicine. However, the discussion in line 338- 341 almost imply that readers should use these antimicrobials (of high importance (amikacin, imipenem and meropenem) to treat bacterial rhinitis in cats. This is an extremely dangerous idea and will further contribute to antimicrobial resistance in both human and veterinary medicine. It is not unsurprising that authors found Pseudomonas spp. being resistant to most antimicrobials apart from these high importance ones, however authors should be cautious when discussing these findings that underlying nasal disease should exclude first by extensive workup before treating bacterial infection and these antimicrobials are antimicrobials of high importance and should not be used easily. Authors should also refer readers to the ISCAID guidelines (Lappin 2017) with use of antimicrobials.
Response 12: Thank you for your suggestion. I have corrected as your suggestion.
Point 13: Reference 33: please use another more appropriate referencing, this is for urinary tract infections.
Response 13: I have removed that as your suggestion.

Reviewer 3 Report
Material and methods:
2.1. Animal experiments and clinical evaluation:
- Sampling collection and management were a bit confusing. Lines 110 -111 “nasal flush samples having significantly higher bacterial cultures” what do you mean with this? How was the quality of the bacterial growths (% of pure and mixed cultures)? How did you register the mixed cultures?
- Please specified the CLSI references used (from human or animal sources?)
Results:
-Remove table 1 and table 2, they are irrelevant. Results of table 1 are already commented in the text (so it is redundant) and the information of table 2 can be write down in the text.
-Table 4. What is the meaning of **?
-Bacterial information from rhinitis cats. Repeated information in lines 227-232 and table 5. Please decide to report it only once (in the text or in the table). Number of isolates can also be included in table 5 (if the authors keep the table).
-Antimicrobial susceptibility: table 4, 5, 6 and 7 should follow numerical order and appear as 6, 7, 8 and 9. However, I recommend changing these tables for bar graphics to show the results in a more visual way.
Discussion:
-Lines290-291: Domestic shorthair (DSH) appeared to be most common breed in this study and also in the cat population in Thailand. And? What is the connection with the previous text??
-rhinitis cats is correct? Maybe the use of cats with rhinitis is more addient…
-Lines 302-304: I cannot see the relationship between the prevalence of rhinitis in younger and adolescent cats and the “allergic” rhinitis in humans.
-Lines325-330: diagnostic difficulties. Why is not performed other techniques such as Ag testing or PCR for the detection of Chlamydophila felis and PCR for mycoplasma spp? I do not agree to the comment of the difficulty of detecting B. bronchiseptica using a selective medium, because with a simple McConkey agar it can be easily isolated.
-Lines 353-355: You can use a more recent paper Li et al., Frontiers in Microbiol 2021 (doi: 10.3389/fmicb.2020.621597) where are reporting similar results of highly susceptibility in Pasteurella spp isolates in cats (from respiratory tract infections).
Conclusions:
It is preferable not to recommend the use of carbapenems in companion animals if you have other antimicrobial choices (following the European guidelines for the use of antibiotics of category D and C in veterinary medicine and preserve the use of category A and B for human medicine).
Author Response
Response to reviewer 3 comments
Point 1: Material and methods:
2.1. Animal experiments and clinical evaluation:
Sampling collection and management were a bit confusing. Lines 110 -111 “nasal flush samples having significantly higher bacterial cultures” what do you mean with this?
Response 1: When comparing sample collection method on nasal culture between nasal flush and nasal tissue biopsy, nasal flush samples were found the growth of bacteria more frequently than nasal tissue samples. According to Johnson study in 2008 “Effect of sample collection bacterial methodology on nasal culture results in cats”.
How was the quality of the bacterial growths (% of pure and mixed cultures)? How did you register the mixed cultures?
Response 2: This report was found 74.26% pure culture and 25.74% is mixed culture. One organism which found heavy pure growth would be considered to be pathogenic germ (Reed, 2020). However, all quality of bacterial growth were collected in this report.
Point 3: Please specified the CLSI references used (from human or animal sources?)
Response 3: The reference is from animal in document M31-A3 (CLSI, 2008)
Point 4: Results:
-Remove table 1 and table 2, they are irrelevant. Results of table 1 are already commented in the text (so it is redundant) and the information of table 2 can be write down in the text.
Response 4: Thank you for your suggestion. I have corrected.
Point 5: Table 4. What is the meaning of **?
Response 5: The two stars mean significant when comparing with the reference group. For example, when comparing age range <1-3 years with age range >10 years, age range <1-3 years appear to occur rhinitis more than neoplasia 4.778 times significantly.
Point 6: Bacterial information from rhinitis cats. Repeated information in lines 227-232 and table 5. Please decide to report it only once (in the text or in the table). Number of isolates can also be included in table 5 (if the authors keep the table).
Response 6: Thank you for your suggestion.
Point 7: Antimicrobial susceptibility: table 4, 5, 6 and 7 should follow numerical order and appear as 6, 7, 8 and 9. However, I recommend changing these tables for bar graphics to show the results in a more visual way.
Response 7: I have changed all of them to bar chart.
Point 8: Discussion:
-Lines290-291: Domestic shorthair (DSH) appeared to be most common breed in this study and also in the cat population in Thailand. And? What is the connection with the previous text??
Response8: this just mean DSH are also main breed in Thailand as well. Since it is not relevant, I will cut it off. (line 321)
Point 9: rhinitis cats is correct? Maybe the use of cats with rhinitis is more addient.
Response 9: I have corrected that.
Point 10: Lines 302-304: I cannot see the relationship between the prevalence of rhinitis in younger and adolescent cats and the “allergic” rhinitis in humans.
Response 10: In human, allergic rhinitis is likely in younger and adolescent and decreasing with age with unclear reason according to the odd ratio in Table 4 and 8 in the study “Epidemiology of allergic rhinitis and associated risk factors in Asia” Chong&Chew, 2018. This result is similar to the current study which found rhinitis is tend to occur in younger and adolescent cats as well.
Point 11: Lines325-330: diagnostic difficulties. Why is not performed other techniques such as Ag testing or PCR for the detection of Chlamydophila felis and PCR for mycoplasma spp? I do not agree to the comment of the difficulty of detecting B. bronchiseptica using a selective medium, because with a simple McConkey agar it can be easily isolated.
Response 11: Absolutely right. That PCR is diagnostic choice for C. felis. However, our study is retrospective study and most cats with rhinitis that came to this hospital either had prolong previous antibiotic treatment or without concurrent ocular sign, C. felis infection were underestimated. Additionally, PCR for Mycoplasma spp. is not routine diagnosis in this hospital. Since Mycoplasma spp. is part of commensal organism in respiratory tract and can be detected in both healthy cats and cats with rhinitis. plus, recent study found it is composition of nasal microbiome community that make positive result of PCR is still questioned. B. bronchiseptica grow well on MacConkey agar however, selective media has some advantages to reduce other bacterial growth since culture is less sensitivity for B. bronchiseptica. (Egberink et al., 2009)
Point 12: Lines 353-355: You can use a more recent paper Li et al., Frontiers in Microbiol 2021 (doi: 10.3389/fmicb.2020.621597) where are reporting similar results of highly susceptibility in Pasteurella spp isolates in cats (from respiratory tract infections).
Response 12: Thank you for your suggestion. I have added this report already.
Point 13: Conclusions:
It is preferable not to recommend the use of carbapenems in companion animals if you have other antimicrobial choices (following the European guidelines for the use of antibiotics of category D and C in veterinary medicine and preserve the use of category A and B for human medicine).
Response 13: I totally agree with your concern and hoping my study could make the animal health care realize AMR situation and the use of antimicrobials.

Round 2
Reviewer 2 Report
Thank you for the changes.
Minor grammatical errors:
Line 63: "was frequently found in mixed culture of.."
Line 64: "an organism that was found with heavy pure growth would be considered... "
Line 106 : "cats with rhinitis that had not had bacterial identification..."
Line 388 "current"
Line 389 "unlike results have been reported in other studies which showed..."
Line 391: "Data including the minimal inhibitory concentration and anaerobic bacteria is not available because..."
Author Response
Department of Companion Animal Clinical Sciences,
Faculty of Veterinary Medicine
Kasetsart University, Thailand
June 8, 2022
Dear Editors and Reviewers;
We appreciated your advice and improvement of our manuscript. In the second rounds revision, we have marked up by using the track changes as detailed below.
Point 1: Include in the text the information of Response 2: "this report was found 74.26% pure culture and 25.74% is mixed culture."
Response 1: I have added this text in line 199-200.
Point 2: Include in the text and references Point 3: Response 3: The reference is from animal in document M31-A3 (CLSI, 2008)
Response 2: I have included this text in line 114-115.
Point 3: Correct grammatical errors
Line 63: "was frequently found in mixed culture of.."
Line 64: "an organism that was found with heavy pure growth would be considered... "
Line 106: "cats with rhinitis that had not had bacterial identification..."
Line 388 "current"
Line 389 "unlike results have been reported in other studies which showed..."
Line 391: "Data including the minimal inhibitory concentration and anaerobic bacteria is not available because..."
Response 3: I have corrected all the grammar error in line 63, 64, 106, 390-392, 393-396.
Yours sincerely,
Jatuporn Rattanasrisomporn, DVM., Ph.D.
Corresponding author
Email: [email protected]

Reviewer 3 Report
Minor changes:
-Include in the text the information of Response 2: "this report was found 74.26% pure culture and 25.74% is mixed culture."
-Include in the text and references Point 3: Response 3: The reference is from animal in document M31-A3 (CLSI, 2008)
Author Response

(The authors gave the same response as above.)
